# Stacked Siamese Generative Adversarial Nets: A Novel Way to Enlarge Image Dataset

Shanlin Liu [1], Ren Han [1,*] and Rami Yared [2]

1. School of Optical-Electrical and Computer Engineering, University of Shanghai for Science and Technology, Shanghai 200093, China
2. Department of Computer Science, Arab International University, Ghabaghib 16180, Syria
* Correspondence: ren.han@usst.edu.com

**Abstract:** Deep neural networks often need to be trained with a large number of samples in a dataset. When the training samples in a dataset are not enough, the performance of the model will degrade. The Generative Adversarial Network (GAN) is considered to be effective at generating samples, and thus, at expanding the datasets. Consequently, in this paper, we proposed a novel method, called the Stacked Siamese Generative Adversarial Network (SSGAN), for generating large-scale images with high quality. The SSGAN is made of a Color Mean Segmentation Encoder (CMS-Encoder) and several Siamese Generative Adversarial Networks (SGAN). The CMS-Encoder extracts features from images using a clustering-based method. Therefore, the CMS-Encoder does not need to be trained and its output has a high interpretability of human visuals. The proposed Siamese Generative Adversarial Network (SGAN) controls the category of generated samples while guaranteeing diversity by introducing a supervisor to the WGAN. The SSGAN progressively learns features in the feature pyramid. We compare the Fréchet Inception Distance (FID) of generated samples of the SSGAN with previous works on four datasets. The result shows that our method outperforms the previous works. In addition, we trained the SSGAN on the CelebA dataset, which consists of cropped images with a size of $128 \times 128$. The good visual effect further proves the outstanding performance of our method in generating large-scale images.

**Keywords:** Generative Adversarial Network; superpixel; SLIC; neural network; data enhancement

## 1. Introduction

The deep neural network is considered to be a powerful model that has wide applications in many fields, such as image classification, object detection, and image segmentation. Gao Y et al. [1] proposed a novel deep learning architecture called the recurrent 3D convolutional neural network. For an effective model, there are three key conditions: a well-designed structure, a large amount of training data and a powerful computing resource. We can strengthen computing resources with the help of hardware facilities and obtain excellent model structures through experience and skill. However, in some scenarios, it is impossible to obtain a large number of training samples. Especially in the field of computer vision, there exists high requirements for the authenticity and validity of images, which leads to the deficit of training data. Lacking valid training data often creates problems such as overfitting, which may greatly degrade the generalization performance of the model.

Data augmentation techniques can be used to augment existing datasets. There are two main types of data enhancement techniques for image data: One is to implement some known change with the existing data, such as panning, rotating, cropping, and flipping. This kind of method can be seen as exerting some subtle dithering on the data distribution. Another way is to generate data in accordance with the distribution of the training dataset, using a generative model such as the Generative Adversarial Network [2]. Although much research has been conducted, it is still challenging to generate large-scale images with high

quality. This paper proposes a novel Generative Adversarial Network named the Siamese Generative Adversarial Network (SGAN), where a pretrained supervisor is introduced to control the category of generated images while guaranteeing diversity. This paper also proposed a new training method for generating large-scale images by progressively generating features in the feature pyramid, which is extracted by the proposed CMS-Encoder. Our contributions are as follows:

- SGAN exerts an additional restriction on the generator by introducing auxiliary loss to the generator. As a result, the generator is expanded to a conditional model and the generator can be effectively guided by the conditional information.
- We propose a feature extraction method based on a clustering algorithm, named Color Mean Segmentation (CMS), which can reduce complexity but still reserve the main structure and key characteristics of image data. We use CMS to construct the CMS-Encoder, which can extract abstract features from images. Compared to the convolutional neural network methods, the CMS-Encoder has three advantages: (1) Since CMS only makes use of the information of the input image, it does not need to be trained. (2) Features extracted by the CMS-Encoder have high interpretability of visuals. (3) The abstract level of output can be controlled by simply adjusting the number of superpixels.
- We proposed a new training method for generating samples. Several SGANs are stacked together to build a Stacked SGAN (SSGAN). By progressively learning features, the SSGAN can learn the distribution of the original dataset.

The paper is organized as follows: In Section 2, we summarize the previous works on Generative Adversarial Networks, as well as the key methods used in our model. In Section 3, we elaborate on the theoretical foundation of our proposed model and algorithm. In Section 4, we discuss the details of our experiments. We compare four auxiliary losses on two simple datasets to determine their effect on our model's training and we demonstrate the leading effect on the generator. Additionally, we demonstrate the ability of the SSGAN by visualizing different levels of a feature map generated by its generators at different levels. In Section 5, we discuss the experiments. In Section 6, we draw conclusions about our work.

## 2. Related Work

In this section, we briefly introduce the Generative Adversarial Networks, Siamese Network and the Simple Linear Iterative Clustering algorithm that will be used in the proposed method.

### 2.1. Generative Adversarial Networks

The Generative Adversarial Network was first proposed by Goodfellow et al. [2], and consisted of a generator and a discriminator. The generator generates fake samples, the discriminator distinguishes between the real samples and fake ones, and together, they play a minimax game until reaching the Nash equilibrium. At that time, the generator can generate samples which have a consistent distribution that matches the original dataset. Jolicoeur-Martineau, A. et al. [3] believed that the probability value of the real samples should become bigger gradually and the probability value of the generated samples should become smaller gradually during training. In [4], a greedy-based winner recruitment strategy was proposed to achieve intelligent information control with maximum credibility and cost. The other notorious problem of the Generative Adversarial Network is "mode collapse" [5], that is, the diversity of the generated samples is poor. In some extreme cases, the generator only generates samples of a single category. Arjovsky, M. et al. [6] convinced that their proposed Wasserstein distance can be used for a generator's gradient update. Additionally, it is proved that the Wasserstein GAN could effectively prevent the disappearance of the generator's gradients, and it does not need to carefully maintain the balance between generator and discriminator. Owing to its outstanding properties, the Wasserstein GAN has a wide application in many fields. Zhang, A. et al. [7] proposed a

multi-generator conditional Wasserstein GAN method to generate a high-quality artificial EEG signal. Yin, Z. et al. [8] proposed a Generative Adversarial Network, which combines multi-perceptual loss and fidelity loss on the basis of the Wasserstein GAN, to generate samples for low-dose computed tomography (LDCT). Gulrajani, I. et al. [9] proposed to add a penalty item to a critic's loss function to satisfy the Lipschitz condition, and proved it works better than clipping the value of a critic's parameters. Gurumurthy, S. et al. [10] mapped the distribution of the input noise to a Gaussian distributionto improve the complexity of the input noise's distribution, and proved that the diversity of the generated samples increased due to this approach. Wu, C. et al. [11] proved that deep reinforcement learning can be successfully applied to a game with four directions of movement. The authors of [12] applied the pix2pix Generative Adversarial Network [13] and cycle-consistent GAN [14] models to convert visible videos to infrared videos. The authors of [15] proposed to focus on target areas using an Attention Generative Adversarial Network, which preserves the fidelity of target areas.

To make the generator model more controllable, Mirza, M. et al. [16] expanded the standard GAN to a conditional model. The conditional information is fed to the generator together with the noise. Additionally, the discriminator not only needs to distinguish between the fake and real sample, but also to discern whether they match or not. Fu, A. et al. [17] demonstrated that the shared gradient still has sensitive information. Chen, X. et al. [18] proposed to divide the input vector of the generator into two separate parts, where one part is an incompressible noise vector and the other part is an interpretable hidden variable. The model learns the relationship between generated samples and hidden variables through another neural network. As a result, the traits of generated samples can be controlled by modifying the value of the hidden variable. The work in [19] separately controls the states of RF and a sensing unit based on a dynamic coordinated reconstruction mechanism.

Some existing works have been designed to train networks in a progressive way. Wu, C. et al. [20] adopted the thinking of layer-by-layer training to increase the hidden layer. Krras, T. et al. [21] proposed a novel way for training. At the beginning, the generator only learns to generate small-sized samples, and the size of the target samples is progressively enlarged. Additional layers are added to the generator and discriminator while the target size becomes larger. Denton, E.L. et al. [22] proposed to generate high-resolution samples by constructing a Laplace pyramid. Xia, F. et al. [23] demonstrated that the inflexible First-Come-First-Served (FCFS) GTS allocation policy and passive deallocation mechanism significantly impair network efficiency. Huang, X. et al. [24] used a convolutional neural network as an encoder to extract features from input images, and then used multiple Generative Adversarial Networks to learn these features in order. The authors of [25] divided the underlying platform into two asymmetric partitions. The basic idea of our work is similar to that of Huang's work, but we propose a novel encoder on the basis of a traditional image segmentation algorithm. Compared to the deep neural network, it does not need to be trained, and the output features have high interpretability of visuals.

*2.2. Simple Linear Iterative Clustering*

kNN is a kind of clustering algorithm that has been widely used. Chen, Y. et al. [26] used a fast approximate kNN algorithm to detect core blocks (CBs), noncore blocks, and noise blocks. Simple Linear Iterative Clustering (SLIC) [27] is a segmentation algorithm based on kNN. It segments an image into several units called superpixels. The superpixel contains several pixels, which have similar features such as color, lightness, texture, and spatial position. The superpixels retain valid information for further image segmentation, and generally do not destroy the boundary information of objects in the image. Therefore, this usually serves as the preprocess step in many areas of computer vision. In [28], the sensory data is preprocessed to obtain abstract features.

Compared to the learning methods of today, the prominent advantage of a traditional segmentation algorithm is that it does not need to be trained. Therefore, it suits the scenario where training datasets are scarce. Compared to other traditional superpixel segmentation

algorithms, SLIC has advantages in terms of convergence time, superpixel compactness, and contour retention. Additionally, it can also be used to segment grayscale images.

### 2.3. Siamese Network

A Siamese Network [29] consists of two neural networks that share their weights with each other. The main idea is to learn a function to map the input into another space. In the target space, the similarity between two inputs can be calculated using geometric metrics, such as Euclidean distance. The Siamese Network model classifies objects by comparing their similarity value with samples of each category.

## 3. Stacked Siamese Generative Adversarial Network

### 3.1. Siamese GAN

As mentioned in [30], multitarget learning can benefit the model, including its convergence and generalization. Zhao, J. et al. [31] introduced indirect trust to strengthen the interaction information when the uncertainty of direct trust is high enough. Enlightened by this, we propose a novel Generative Adversarial Network called the Siamese GAN, which is based on the Wasserstein GAN with a gradient penalty (WGAN-GP) [9]. To control the category of the generated samples, we expanded the generator of the WGAN to a conditional model. The generator was fed with a noise vector and combined with the conditional information. A new object was added to the generator to expand the generator's training to a multi-object task. We trained the Siamese Network, which serves as a supervisor in our model, to discern the similarity between two samples. One input of the supervisor is for samples generated by the generator of the WGAN, and the other is for samples from the training dataset, which contains the corresponding category to the generator's conditional information. The output of the supervisor is the similarity value of the two inputs, and it is used as the auxiliary loss of the generator. By minimizing the auxiliary loss, the generator can generate samples corresponding to the conditional information. The other three similarity metrics are also used as the auxiliary loss, and we compare their effect on the generator model in Section 4. We use the WGAN as the basic model due to its stability. The introduction of the auxiliary loss breaks the balance between generator and discriminator, and it takes several epochs to return to the balance status again. Using the WGAN does not require us to focus on the balance problem. We use the Siamese Network as the supervisor because it does no harm to the diversity of generated samples. The structure of the Siamese GAN is shown in Figure 1b and the structure of the vanilla GAN is shown in Figure 1a as a comparison.

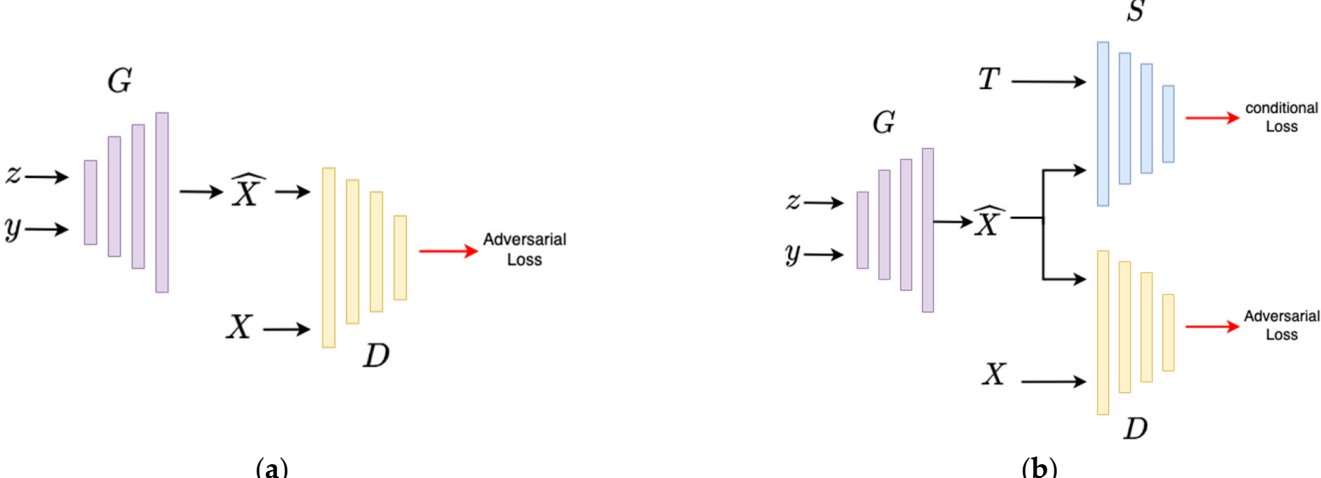

**Figure 1.** Vanilla GAN and Siamese GAN. (**a**) Vanilla GAN. (**b**) Siamese GAN.

In Figure 1, *G* and *D* represent the generator and discriminator, respectively. *S* is the introduced supervisor, *y* is conditional information such as category labels, *X* stands for the training samples from the original dataset, $\hat{X}$ stands for samples generated by the generator, and *T* stands for the samples that will be sent to the supervisor together with the generated samples. Compared to the original GAN, the update of our proposed generator's gradients not only depends on the backpropagation of the discriminator *D*, but also depends on the backpropagation of the supervisor *S*. As a result, the generated samples will not only have fidelity, but will also be similar to the samples of *T*. By exerting such constraints on the generator, its training process will be effectively guided by conditional information.

The loss function of the generator of the SGAN is expressed as

$$L_G = \omega_1 L_G^{adv} + \omega_2 L_G^{cond} \tag{1}$$

which consists of two parts: adversarial loss $L_G^{adv}$ and conditional loss $L_G^{cond}$. The adversarial loss reflects the generated samples' fidelity to the original samples. The conditional loss serves as auxiliary loss, which reflects the match degree between the generated sample and conditional information. The total loss of the generator is expressed as a linear weighted sum of such two losses. $\omega_1$ and $\omega_2$ are two hyperparameters. If the value of $\omega_1$ is fixed, as the value of $\omega_2$ increases, the generated samples will have higher fidelity but will be less diverse. Conversely, as the value of $\omega_2/\omega_1$ decreases, the generated samples will become more diverse but will have less fidelity. When $\omega_2 \approx 0$, the SGAN degrades to the WGAN because the conditional loss has little effect.

In Equation (1), the adversarial loss $L_G^{adv}$ is the same as that defined in [9]. The conditional loss $L_G^{cond}$ is represented as

$$f(\hat{X}, T) = \|\hat{X} - T\|_2, \tag{2}$$

where $\hat{X}$ stands for the generated samples and *T* stands for samples sent to the supervisor together with $\hat{X}$.

As mentioned in Section 2.3, The Siamese Network calculates the similarity between two samples by mapping them into another space. If the function $h(x)$ fits the process of mapping, then their similarity can be calculated by using the Euclidean distance, which is expressed as

$$f(\hat{X}, T) = \|h(\hat{X}) - h(T)\|_2. \tag{3}$$

*3.2. CMS-Encoder*

Based on [32], we encode images via the proposed CMS-Encoder. Considering the factors that affect the complexity of the image distribution, there are two main points: the complexity of the image pixels' distribution and image size. Therefore, we consider abstracting the images from these two aspects.

To simplify the distribution of image pixels, firstly, we segment input images into superpixel-level images, which are composed of multiple superpixels, by using the SLIC superpixel segmentation algorithm. Since all pixels within each superpixel have similar attributes such as color and texture, the color of each superpixel is set to the average value of every pixel within it. By doing this, the input image is transformed from a pixel-level image to a superpixel-level image, and the distribution complexity of input data is reduced heavily. We define the operation above as the Color Mean Segmentation (CMS), which is the core of our proposed CMS-Encoder. The details of the CMS algorithm are presented in Algorithm 1.

---

**Algorithm 1** Color Mean Segmentation

---

/* segmentation*/
Initialize cluster centers $C_k = [l_k, a_k, b_k, x, y_k]^T$ by sampling pixels at regular grid steps S.
Move cluster centers to the lowest gradient position in a $3 \times 3$ neighborhood.
Set label $l(i) = -1$ for each pixel i.
Set distance $d(i) = \infty$ for each pixel i.
Set color $cl(i) = (0, 0, 0)$ for each pixel i.
**repeat**
    **for** each cluster center $C_k$ **do**
        **for** each pixel i in a $2S \times 2S$ region around $C_k$ **do**
            Compute the distance D between $C_k$ and i.
            **if** $D < d(i)$ **then**
                set $d(i) = D$
                set $l(i) = k$
            **end if**
        **end for**
    **end for**
    Compute new cluster centers.
    Compute residual error E.
**Util** $E <= $ threshold
/* Mean */
**for** each cluster u **do**
    compute mean red color $r_u$ of every pixel in cluster u.
    compute mean green color $g_u$ of every pixel in cluster u.
    compute mean blue color $b_u$ of every pixel in cluster u.
    **for** each pixel i in cluster u **do** :
        set $cl(i) = (r_u, g_u, b_u)$
    **end for**
**end for**

---

In order to scale the size, the input image needs to be downsampled. We selected the maxpooling operation because it preserves the local features of the image. Thus, the maxpooling operation is one of the layers of our proposed CMS-Encoder, and the other one is the CMS operation mentioned above. We chose to place the maxpooling layer before the CMS layer in order to preserve more important information, such as points on edges. By combining these two layers together, we obtained the single unit of the CMS-Encoder, which is shown in Figure 2. It consists of two layers: the maxpooling layer and CMS layer. They work sequentially to abstract input images.

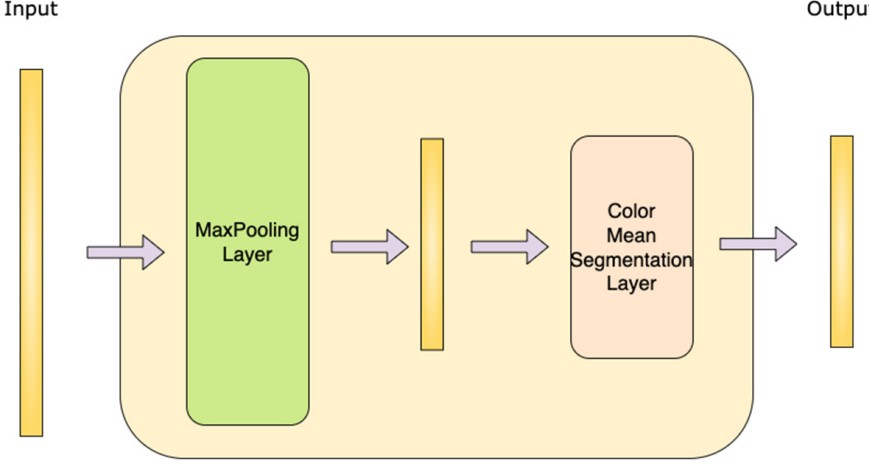

**Figure 2.** Structure of single unit of CMS-Encoder.

When stacking several such units together, we obtained our CMS-Encoder, which can extract different levels of abstract features of the input image; thus, the size of the features gradually becomes smaller and the distribution of the features gradually becomes simpler. However, they all preserve the main characteristics of the original image. When we put these features together from bottom to top, it is similar to a feature pyramid. The topmost feature has the smallest size and the simplest distribution. As the position moves down, the size of the features becomes larger and larger and the distribution becomes more and more complex. The original image is located at the bottom of the pyramid. Figure 3a shows the workflow of a two-unit CMS-Encoder. The abstraction degree of the image becomes higher and higher, and the distribution becomes simpler and simpler, but the outputs of every unit still preserve the main structure and key characteristics of the original image. Figure 3b shows the produced feature pyramid by putting the output features in a certain order.

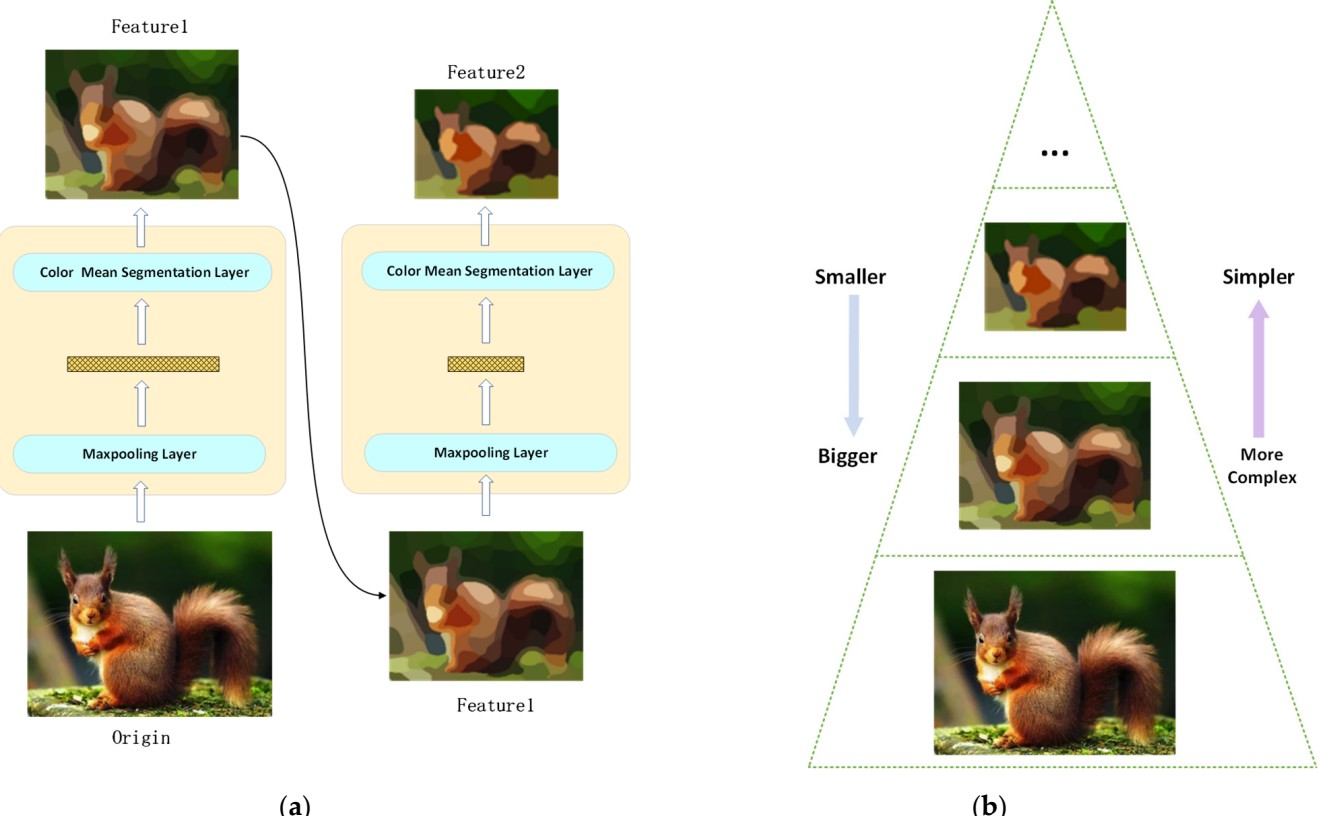

**(a)**      **(b)**

**Figure 3.** CMS-Encoder with two units. (**a**) The workflow of a CMS-Encoder with two units. (**b**) The "feature pyramid" extracted by the two-unit CMS-Encoder.

The proposed CMS-Encoder has three significant advantages over the convolutional neural network (CNN):

(1) Do not need to train. The CMS-Encoder alleviates the dependency on the training process and training dataset, which is critical for the CNN model. Since the CMS algorithm is based on the clustering algorithm, all the computations only rely on the input image itself. Therefore, it does not need any cautious designs for training, and we do not need to worry about the dataset for its learning.

(2) Highly interpretable. The CNN is famous for its ability of extracting features of images, but it is also known as a black-box model. When we use the CNN as an end-to-end model, we do not understand what happens in it due to its terrible interpretability. However, for our proposed CMS-Encoder, we can visualize features and readily observe what they represent from a human perspective.

(3) Controllable. The CMS layer, which acts as the core in the CMS-Encoder, is based on the SLIC algorithm. Therefore, we could control the number of superpixels to be segmented. As we mentioned above, the output feature can be seen as a superpixel-level image, and the number of superpixels directly decides the abstract degree of the feature. Hence, the abstract level of the feature can by controlled by simply changing the number of superpixels.

### 3.3. Stacked Siamese GAN

By using the CMS-Encoder, we can extract a feature pyramid, which consists of several features extracted from the original image. From the top of the feature pyramid to the bottom, the size of the feature becomes bigger and bigger, and the distribution becomes more and more complex. When we move from top to bottom, it seems as if someone scales up the feature and then fills in more details. That is the main idea of our proposed SSGAN. We use multiple SGANs to work sequentially. At the beginning, the first SGAN is responsible for generating the most abstract features from the noise vector and conditional information. Then, the generated feature is sent to the next SGAN as conditional input. The next SGAN scales up the feature size and fills in the details to generate a more concrete one. In this way, each of the SGANs generates more detailed features from the feature outputs of the previous generator. The feature is passed down, and more and more details are filled in during the passing. Finally, the last generator can generate samples with the same distribution as the original images. The main idea is that we divide the process of learning into several steps, where each step only needs to learn to fill in more details about the input features instead of learning the structure of the input features from scratch. Figure 4 shows the architecture and workflow of the SSGAN and Table 1 explains the meaning of each symbol.

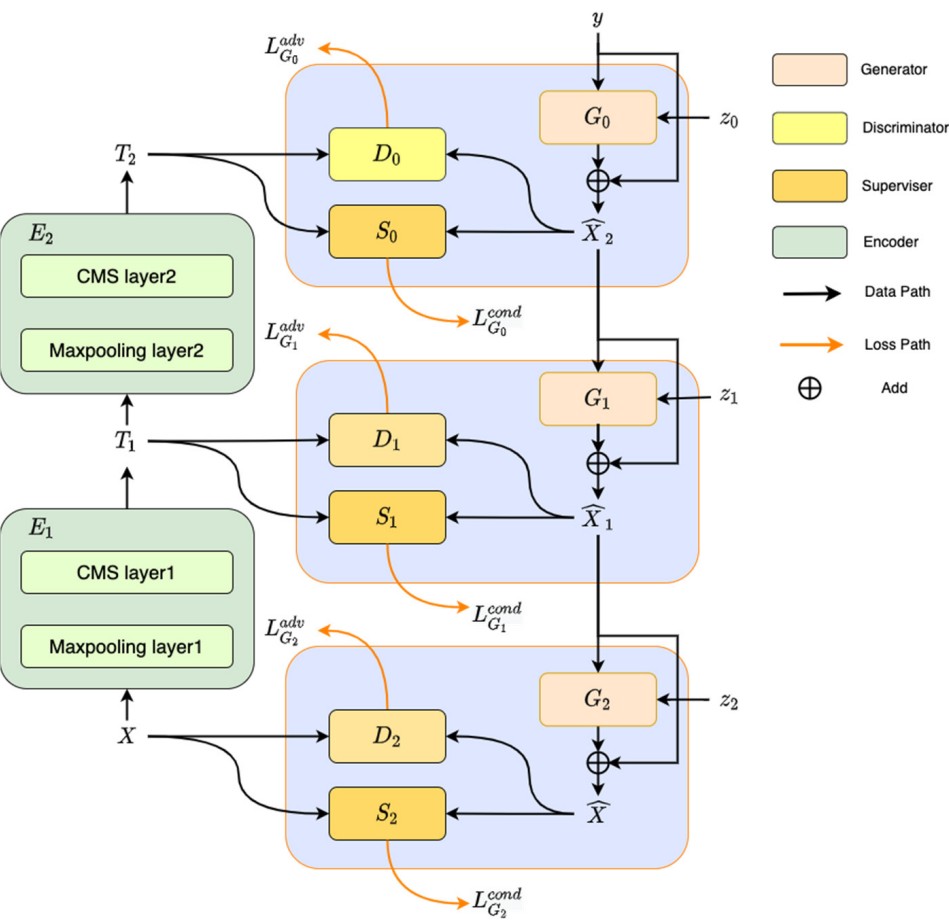

**Figure 4.** The structure of Stacked Siamese Generative Adversarial Network.

**Table 1.** Instruction for the symbols in Figure 4.

| Symbol | Meaning |
| --- | --- |
| $G$ | Generator |
| $D$ | Discriminator |
| $E$ | Encoder |
| $S$ | Supervisor |
| $X$ | Training sample |
| $\hat{X}$ | Generated sample |
| $y$ | Label |
| $T$ | Extracted feature |
| $\hat{T}$ | Generated feature |
| $L_G^{adv}$ | Adversarial loss |
| $L_G^{cond}$ | Conditional loss |

The architecture of each generator $G$ is inspired by ResNet [33]. One of the inputs of $G$ is the noise vector randomly sampled, and the other is the conditional information. The generator is composed of two paths: the main path and identity path. The main path follows the structure of the DCGAN. On one hand, the conditional information is sent to the main path. On the other hand, the conditional information is sent to the identity path to be upsampled, which is used to scale the conditional information to match its size with the output of the main path. The identity path is used to maintain the learned feature of the conditional information as mentioned before, and the main path is used to learn more details about the transformation. At the end of the generator, the output of main path and identity path are added together as the final output. For small-scale image datasets, the supervisor can be constructed with a few fully connected layers combined with dropout layers. For more complex datasets, the supervisor may use several convolutional layers combined with maxpooling layers; they mainly work to extract features, which are used to calculate the distance among different samples.

Firstly, the noise vector z and condtional information y are sent to the first generator $G_0$, and its gradients are updated by the backpropagation of both adversarial loss $L_{G_0}^{adv}$ and conditional loss $L_{G_0}^{cond}$. In order not to shock the model, once the training of $G_0$ converges, layers of generator $G_0$ need to be frozen. Then, $G_0$'s output $\hat{X}_2$ is used as the conditional information of the next generator $G_1$. After the training converges, layers of $G_0$ are unfrozen, and $G_0$ and $G_1$'s combination is finetuned. Then, $G_0$ and $G_1$ are frozen again to train $G_2$, and the same thing will happen to every following generator.

## 4. Result Analysis

This section begins with the introduction of four datasets: MNIST [34], Fashion-MNIST [35], Cifar-10 [36], and CelebA [37]. We first pretrain the Siamese Network, and then we use four similarity metrics as the auxiliary conditional loss of the SGAN's generator, including the cosine distance, mean squared error, output of the Siamese Network (SGAN), and cross-entropy value of a trained classifier, to compare their effect on the training of the model. Finally, we show the ability of the proposed SSGAN to generate large-scale images from a CelebA dataset, and we visualize the produced features of each generator through the SSGAN.

### 4.1. Introduction to the Dataset

MNIST [34] is a classic dataset in the field of machine learning, which contains handwritten digits in 10 categories from 0 to 9. The whole dataset is divided into a training set and test set. The training set contains 60,000 images and corresponding labels, and the test set contains 10,000 images and corresponding labels. Both the height and width of the images are 28. MNIST is widely used in the field of machine learning for various tasks to detect the ability of models.

Fashion-MNIST [35] covers frontal images of 70,000 different products in 10 categories: T-shirts, jeans, pullovers, skirts, coats, sandals, shirts, sneakers, bags, and boots. The division of the test set/training set is the same as that of the MNIST, and its image size is $28 \times 28$.

CIFAR-10 [36] contains 50,000 training samples and 10,000 test samples, where each sample is a three-channel colored image with a size of $32 \times 32$, and the dataset contains different natural scenes from 10 categories.

The CelebFaces Attributes Dataset (CelebA) is a large-scale face attribute dataset with more than 200K celebrity images, each with 40 attribute annotations. The images in this dataset cover a large variety of poses and background clutter. CelebA has a large diversity, large quantities, and rich annotations [37].

### 4.2. Guiding Effect of Siamese GAN

We implement the supervisor as a three-layer perceptron, and a dropout layer is added between every two fully connected layers. 60,000 samples from the MNIST are used to build the training set and 10,000 samples are used to build the test set. We use the RMSProp [38] as the optimizer and set the training rate as $1 \times 10^{-4}$. The training batch size is set to 128.

Figure 5a,b shows the trend of loss and accuracy during training, respectively. During the first ten epochs, the loss drops quickly, and gradually converges after 40 epochs. At that point, the accuracy of the model reaches 99.49% on the training set and 97.83% on the test set.

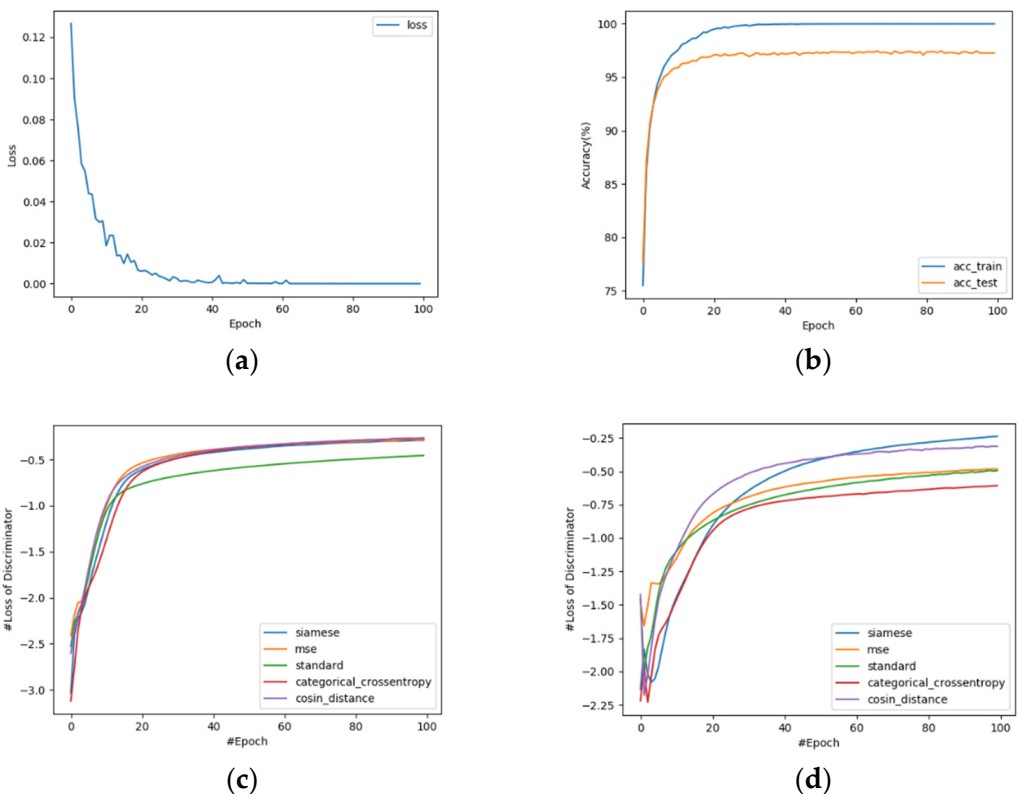

**Figure 5.** Training Siamese Network on MNIST dataset. (**a**) The trend of loss as epoch increases. (**b**) The trend of accuracy on training set and test set as epoch increases. (**c**) Trend of discriminator's loss trained with different conditional losses on MNIST. (**d**) Trend of discriminator's loss trained with different conditional losses on Fashion-MNIST.

We then use four similarity metrics as the auxiliary conditional loss of the SGAN and compare their effect on the generator's training. Besides the Siamese Network, we train a classifier and use its result to calculate the categorical cross-entropy loss. Cosine distance and mean squared error are also used as metrics of two images' similarity. The

generator and discriminator are constructed as the DCGAN [39]. Since the loss of the WGAN's discriminator reflects the quality of generated samples [6], we show how the loss of the discriminator changes with different auxiliary losses on the MNIST and Fashion-MNIST datasets in Figure 5c,d, respectively. When training on a simpler dataset, all four losses improve the quality of generated samples. However, when the dataset becomes more complex, a model with cross-entropy loss does not work as well as before, and the quality of its generated samples becomes even worse than the original WGAN. It is probably because the bad backpropagation occurs on the classifier. The model trained with output from the Siamese Network still works well and converges better than others in the latter epochs. Figure 6a,b presents the conditional information's guiding effect on the SGAN's generator.

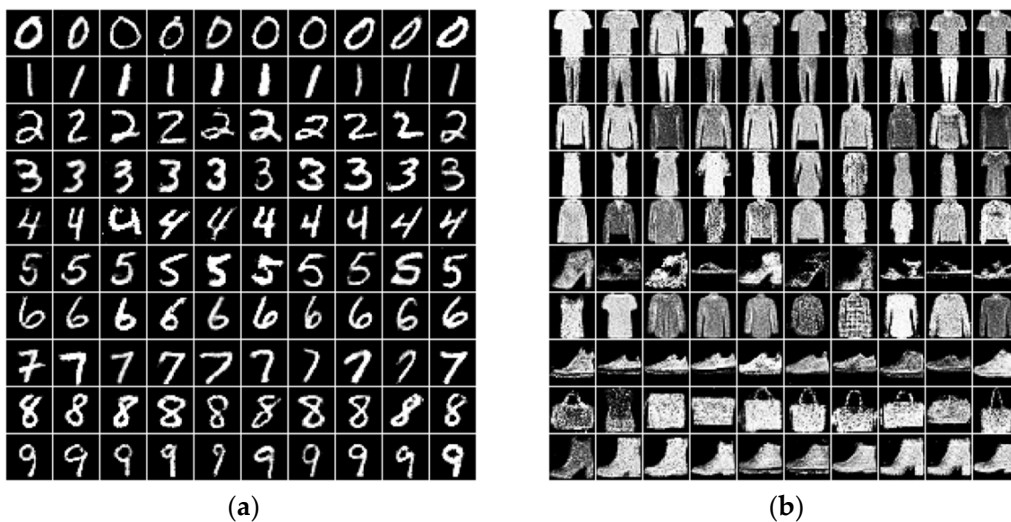

(**a**) (**b**)

**Figure 6.** Samples generated by SGAN. (**a**) MNIST. (**b**) Fashion-MNIST.

In addition, we further study the impact of the pretrained supervisor's accuracy on the convergence of the discriminator. The architecture of the generator, discriminator, and supervisor are the same as above. We pretrain four supervisors to reach an accuracy of about 70%, 80%, 90%, and 99%. Figure 7 shows the convergence of the discriminator changes as the accuracy of the pretrained supervisor changes. It can be found that as the accuracy of the pretrained supervisor goes down, the training of the corresponding discriminator converges to a lower point, which means that the quality of generated samples becomes poor. This may be attributed to the poor judgement of the supervisor, which interferes with the parameter update of the generator.

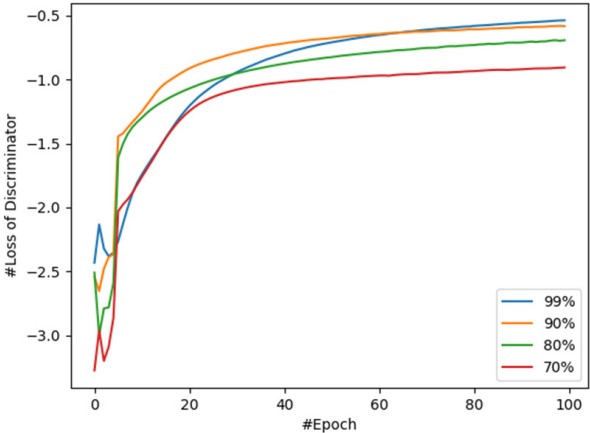

**Figure 7.** Convergence of discriminator changes as the accuracy of pretrained supervisor changes.

### 4.3. Generated Samples with Stacked Siamese GAN

To demonstrate the performance of the SSGAN, we first train the SSGAN on the CelebA dataset. All images in the dataset are cropped from the center to a size of $128 \times 128$. The SSGAN is constructed by three SGANs and two CMS-Encoder units, where the main path of the generator and discriminator of each SGAN have the same structure as the DCGAN [39], and the identity path of each generator is implemented through upsampling with the nearest interpolation. The first CMS-Encoder unit segments 200 superpixels for each image to extract features at the first level. The extracted features are then segmented by the second CMS-Encoder unit to 100 superpixels to generate features at the second level. Figure 8 shows the original samples and intermediate features extracted by the CMS-Encoder from the original samples. Features and samples generated by the SMGAN are also presented in Figure 8. Then, we compare the quality of samples generated by the SSGAN with the results of previous works by using the Fréchet Inception Distance (FID) [40] on four datasets. A lower FID value means the generated samples have higher fidelity and diversity. Table 2 shows that the FID results of our method outperform the results of the state-of-the-art methods.

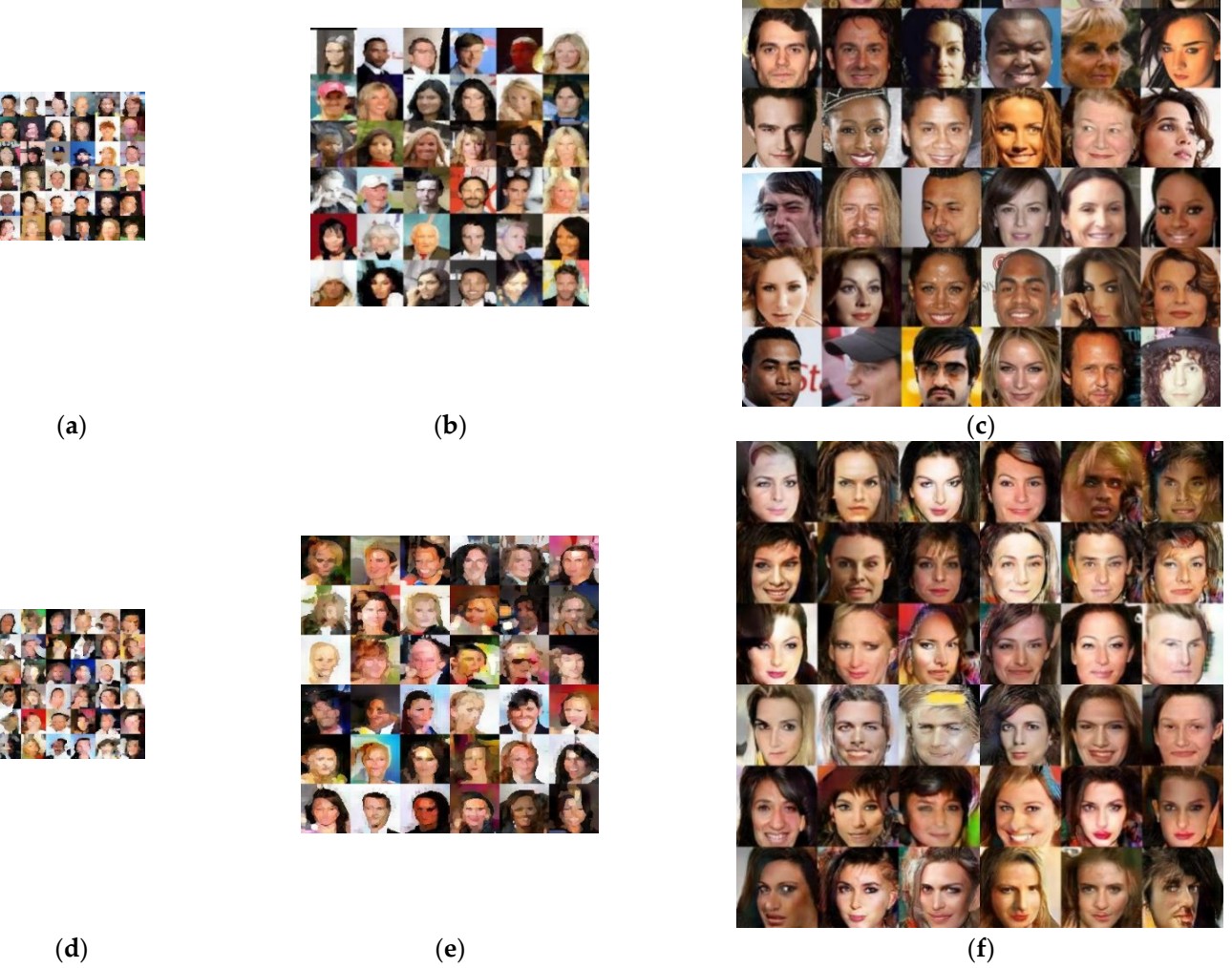

**Figure 8.** Generated and original samples. (**a**) Features extracted by the second CMS-Encoder unit; size: $32 \times 32$. (**b**) Features extracted by the first CMS-Encoder unit; size: $64 \times 64$. (**c**) Original samples; size: $128 \times 128$. (**d**) Features generated by the first SGAN; size: $32 \times 32$. (**e**) Features generated by the second SGAN; size: $64 \times 64$. (**f**) Samples generated by the third SGAN; size: $128 \times 128$.

**Table 2.** FID of SSGAN and other methods. The results show that the proposed SSGAN outperforms previous works.

|  | MNIST | Fashion-MNIST | Cifar-10 | CelebA |
|---|---|---|---|---|
| DCGAN [39] | 19.18 | 23.33 | **27.45** | 17.28 |
| WGAN [6] | 14.27 | 26.43 | 35.37 | 15.53 |
| WGAN-gp [9] | 13.11 | 25.19 | 30.96 | 16.66 |
| LSGAN [41] | 23.40 | 31.76 | 41.42 | 15.75 |
| SSGAN (our method) | **10.94** | **19.98** | 28.38 | **10.34** |

## 5. Discussion

The Siamese Network, used as a supervisor in the SGAN, is valid for the convergence of the model's training. In comparison with other similarity metrics, the Siamese Network still works better when the training dataset becomes more complex. The generated samples show that the auxiliary conditional loss has a good guiding effect on the generation of samples. However, we found that the performance of the pretrained supervisor influences the training of the SGAN and the convergence of the discriminator. We showed the samples generated by the SSGAN, which was trained on the CelebA dataset. Samples in the datasets were cropped to a size of $128 \times 128$. The good visual effect proves the performance of our method. In addition, we compared the FID of the SSGAN and previous works, which further proves the good quality of samples obtained by the SSGAN quantitively. From the experiments, we can conclude that the SSGAN can be applied to more challenging datasets, which have a larger size and more complex distribution.

## 6. Conclusions

We introduced a supervisor to the WGAN, which exerts an auxiliary conditional loss on the generator. The WGAN was, therefore, expanded to a conditional model. It was proved that the auxiliary conditional loss contributes to the convergence of the model. We showed that the SGAN helps with the convergence of the model, and it still works well even as the training dataset becomes more complex. We further found that lower accuracy leads to the poor convergence of the discriminator. The CMS-Encoder abstracts features from images but still maintains their main structure and key characteristics. Moreover, it does not require training and its output has high interpretability.

The SSGAN is composed of a CMS-Encoder and several SGANs. SGANs progressively learn the features in the feature pyramid, which is extracted by the CMS-Encoder. We compared the FID of samples generated by the SSGAN with previous works on four datasets. The results showed that our method outperforms the previous works. We also trained an SSGAN on the CelebA dataset, in which the images were cropped and resized to a size of $128 \times 128$. The good effect of the generated samples demonstrates the outstanding performance of our method. We conclude that the SSGAN has advantages in generating large-scale image samples.

In the future, we intend to study the effect of the SSGAN on datasets with a higher dimension, such as hyperspectral images. We would like to find a brand-new operation to replace the convolutional layers in the generators. The proposed feature pyramid, which is extracted by the CMS-Encoder, can also be used to train a stacked autoencoder. In addition, since our proposed SSGAN can generate features with different granularities, it can be used in the field of anomaly detection or anomaly localization as well, since people tend to localize the anomaly areas with unsupervised learning methods.

**Author Contributions:** Conceptualization, S.L.; data curation, S.L.; formal analysis, S.L.; methodology, S.L.; resources, S.L.; software, S.L.; supervision, R.H.; validation, R.Y.; visualization, R.Y.; writing—original draft, S.L.; writing—review and editing, R.H. All authors have read and agreed to the published version of the manuscript.

**Funding:** This research received no external funding.

**Data Availability Statement:** http://yann.lecun.com/exdb/mnist/; https://github.com/zalandoresearch/fashion-mnist/; https://www.cs.toronto.edu/~kriz/cifar.html; http://mmlab.ie.cuhk.edu.hk/projects/CelebA.html (accessed on 1 December 2022).

**Conflicts of Interest:** The authors declare no conflict of interest.

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
