# Peer review of "Stacked Siamese Generative Adversarial Nets: A Novel Way to Enlarge Image Dataset"

_electronics, doi:10.3390/electronics12030654_

Round 1
Reviewer 1 Report
Thank you for the hard work! The reviewer would suggest some minor editions as below:
1. Explain why your new SGAN is important or what is the need for this in the introduction session.
2. Add some future work (e.g. optimizations).
3. Explain if there are any broader applications.
Reviewer 2 Report
1) Some future work is useful and important in Section 6.
2 Reformat this Figure 7. It its out of this range.
3) Please re-write this abstract to high light the main contributions in this paper.
4) After each function, please add a punctuation.
5) Some recent references are useful for this paper, for example,
[2] Y Zhou, Y Zhang, et al., A bare-metal and asymmetric partitioning approach to client virtualization. IEEE Transactions on Services Computing 7 (1), 40-53, 2012.
[1] Y Zeng, CJ Sreenan, et al., Connectivity and coverage maintenance in wireless sensor networks, The Journal of Supercomputing 52 (1), 23-46, 2010.
Reviewer 3 Report
Please confirm the attached PDF file.

Round 2
Reviewer 3 Report
Thank you for your explanation of my concerns. I understood most of it.
Please correct the minor points listed as below.
1. In Abstract, CMS and FID should be spelled out.
2. In 3.1 section, please use italics for variable symbols (G, D, S, T, etc.)
3. In the sentence after Equation 2, please modify χ and x1.
Author Response
Dear Reviewer:
Thank you for your work. We have corrected the mistakes which you listed in the comments.
1. We added the full name of the CMS and the FID in the Abstract.
2. We changed the font of symbols in Section 3.1 to italic.
3. We modified the variable symbols which are used in the sentences after Equation 2.